# C3Po: Cross-View Cross-Modality Correspondence by Pointmap Prediction

**Kuan Wei Huang**[*]
Cornell University

**Brandon Li**
Cornell University

**Bharath Hariharan**
Cornell University

**Noah Snavely**
Cornell University

## Abstract

Geometric models like DUSt3R have shown great advances in understanding the geometry of a scene from pairs of photos. However, they fail when the inputs are from vastly different viewpoints (e.g., aerial vs. ground) or modalities (e.g., photos vs. abstract drawings) compared to what was observed during training. This paper addresses a challenging version of this problem: predicting correspondences between ground-level photos and floor plans. Current datasets for joint photo–floor plan reasoning are limited, either lacking in varying modalities (VIGOR) or lacking in correspondences (WAFFLE). To address these limitations, we introduce a new dataset, C3, created by first reconstructing a number of scenes in 3D from Internet photo collections via structure-from-motion, then manually registering the reconstructions to floor plans gathered from the Internet, from which we can derive correspondences between images and floor plans. C3 contains 90K paired floor plans and photos across 597 scenes with 153M pixel-level correspondences and 85K camera poses. We find that state-of-the-art correspondence models struggle on this task. By training on our new data, we can improve on the best performing method by 34% in RMSE. We also use the predicted correspondences to estimate camera poses and evaluate performance using recall metrics. Lastly, we identify open challenges in cross-modal geometric reasoning that our dataset aims to help address. Our project website is available at: https://c3po-correspondence.github.io/.

## 1 Introduction

A frequent experience when visiting a tourist site is finding our way around with a map. On the face of it, this is a very challenging task. The bird's-eye view offered by the map or floor plan is completely different from the view we see from the ground. Exacerbating this cross-view challenge is the fact that the modality is also different: the floor plan is abstract and has none of the visual features that we see on the ground. Yet, we regularly solve this challenge, perhaps by relying on correspondences between the two views: for instance, we might map the dome we see above us to the clear circle on the map (Figure 1, right). Given that we are able to draw these correspondences, we ask: can we get computer vision models to do the same?

The ability to draw cross-view, cross-modality correspondences between photos and abstract floor plans also has practical utility. For instance, it can allow robots to localize themselves given just a map and a few sparse views. These correspondences can also be an added source of information when solving challenging structure-from-motion (SfM) problems, since they can be used to register cameras to the floor plan and thus to each other. This may be especially useful in cases where sparse views with limited overlap lead to multiple, disconnected reconstructions that cannot be registered to each other, but could be jointly registered to a plan-view image.

---

[*]Corresponding email: kwhuang@cs.cornell.edu

39th Conference on Neural Information Processing Systems (NeurIPS 2025) Track on Datasets and Benchmarks.

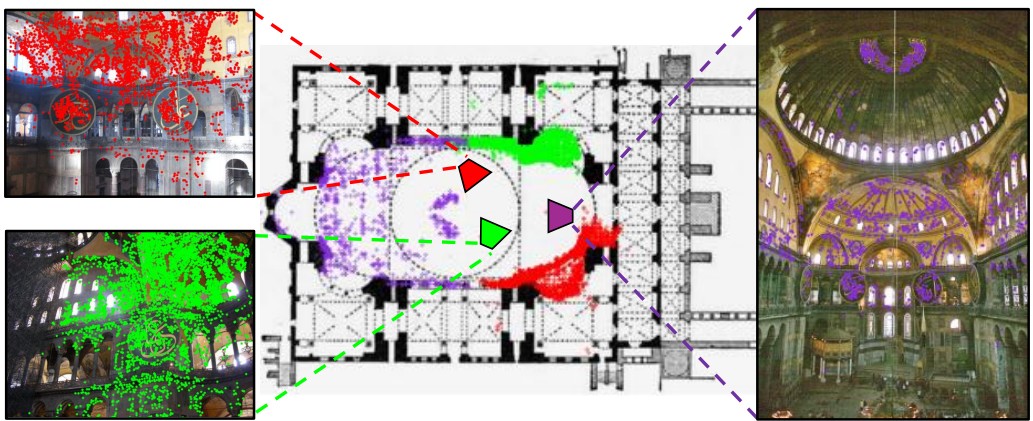

Figure 1: We present C3, a dataset of paired floor plans and photos, annotated with point correspondences and camera poses. It consists of 90K plan-photo pairs, 153M correspondences, and 85K camera poses across 597 scenes, covering a broad range of scene structures, lighting conditions, and camera poses. We show photos captured from various views within an example scene, their camera poses, and color-coded correspondences to the floor plan.

Recent advances in computer vision suggest that current state-of-the-art techniques can indeed identify correspondences even in challenging scenarios. In particular, DUSt3R [1] demonstrated the ability to draw correspondences across "nearly opposite" viewpoints. Self-supervised feature representations like DINO [2] or DIFT [3] have been shown to allow for cross-modality correspondences. Yet these models have never been tested in scenarios that combine the challenge of viewpoint and modality, a challenge exemplified in the image-to-floor plan correspondence problem. The reason is the lack of benchmarking data: there exists no prior dataset that provides ground truth correspondences across images and floor plans.

In this work, we address this gap and create a first-of-its-kind dataset, C3 (for *Cross-View Cross-Modality Correspondence*), of correspondences between floor plans and photos. To address the complexity of manually annotating correspondences, we propose a novel pipeline that uses SfM point clouds manually aligned to floor plans to infer individual point correspondences.

We use our dataset to evaluate state-of-the-art correspondence techniques ranging from DUSt3R to self-supervised features like DINO as well as methods trained explicitly for correspondence [3–7]. We find that existing models struggle to draw these correspondences, with most methods yielding errors that are more than 10% of the image size. This suggests that in spite of large advances in reconstruction, current state-of-the-art is not sufficient for solving the problem of cross-view, cross-modality correspondences.

We improve the state-of-the-art via a method we call cross-view, cross-modality correspondence by pointmap prediction (C3Po) where we fine-tune DUSt3R on our dataset, yielding a 34% reduction in error. Yet, we find that errors are much higher than classical correspondence problems. We analyze the remaining errors and find multiple challenges that are particular to this cross-view, cross-modal problem: often, ground-level photos do not provide enough context of the overall scene, and this problem is exacerbated when symmetries in the structure make the problem ambiguous. Handling this ambiguity is an open problem deserving of future research.

In sum, our contributions are:

1. We present a dataset consisting of floor plan-photo pairs collected from the Internet, along with pixel correspondences for each pair and the camera pose for each photo.

2. We demonstrate that state-of-the-art correspondence techniques fail to draw accurate correspondences between images and floor plans.

3. We adapt DUSt3R's pointmap prediction to estimate correspondences, outperforming the best baseline by 34%.

4. We show camera pose estimation as a downstream application of our predicted correspondences and evaluate its performance with recall metrics.

5. We identify systematic sources of error due to the natural ambiguity in data for future work to explore.

## 2 Related Work

### 2.1 Cross-view and Cross-modal Datasets

Since C3 is, to our knowledge, the first cross-view, cross-domain correspondence dataset, we identify the following three categories of prior datasets that are most relevant to our work: datasets with photos of scenes, floor plans, and cross-viewpoint imagery. Datasets with photos of scenes have been critical in the development of scene understanding methods [8, 9] and training the visual perception for embodied agents [10]. However, they lack scene layout images. Floor plans are a compact and generalizable 2D layout representation of 3D buildings, and prior floor plan datasets have enabled tasks like layout estimation [11, 12], scene generation [13, 14], and others [15–17]. These prior datasets, however, are limited to indoor residential buildings. The recent WAFFLE dataset [18] consists of 20K floor plans covering a wide range of building types and locations, but lacks photos of the corresponding scenes. Finally, cross-view datasets typically contain satellite and ground-level images [19] and have historically been utilized for tasks like geo-localization [20]. While the satellite images provide aerial views, they do not necessarily contain structural information in the abstracted way that floor plans do. They are also missing the correspondences between the images from the two viewpoints. Our dataset not only contains floor plan–photo pairs drawn from a broad array of architectural structures and geographical regions, but also provides 2D correspondences between the pairs and a camera pose for each photo.

### 2.2 Pixel Correspondence

Pixel correspondence, the process of finding matching points in different images that represent the same point in the real world, is a fundamental task in computer vision with a wide range of applications, including 3D reconstruction [21–23], motion tracking [24–26], and geo-localization [27–29]. Historical matching methods rely on manually engineered features, like SIFT [30], SURF [31], and ORB [32]. Newer strategies have shifted towards learning-based methods [33], for their ability to extract more adaptable features automatically from data. However, these features are local, limiting their robustness in capturing broader scene context. To address this, dense methods [4, 5] have been introduced to establish correspondences at a global scale for better performance, especially in extreme conditions, like large viewpoint changes, textureless areas, and poor lighting, but have yet to be tested on inputs from different visual modalities. Recently, DUSt3R [1] demonstrated the ability to learn scene geometry from pairs of photos. At its core, DUSt3R predicts a pointmap which creates a mapping from 2D image pixels to 3D scene points, and we turn this pointmap prediction into a correspondence prediction. While we find that DUSt3R alone is not effective at cross-view, cross-modality matching, we discuss how we adapt it to this task in Section 4.

## 3 C3: Cross-View, Cross-Modality Correspondence Dataset

Our goal is to create a dataset that consists of paired floor plans and photos and annotated correspondences between them. Two key challenges faced in building such a dataset are (1) finding a good source of floor plan images, and (2) determining correspondences between floor plans and corresponding photos. In the following sections, we describe how we address these challenges and produce our dataset.

### 3.1 Sourcing Floor Plans

We source our floor plans from Wikimedia Commons, following past work [18]. Wikimedia Commons is an online media repository that hosts freely licensed media content, including images, sound, and video clips. Data is organized into a hierarchy of categories and subcategories (for instance, *Cathédrale Notre-Dame de Paris → Interior of Cathédrale Notre-Dame de Paris → Plans of Cathédrale Notre-Dame de Paris*). Images and other files form the leaves of the hierarchy and can belong

to multiple categories. Each file includes metadata, including captions, source, and the categories it belongs to.

To collect floor plans from Wikimedia Commons, we start by recursively traversing through the *Floor plans* category and considering every image file in the subtree. We filter these images in the following way. We observe that floor plans are typically tagged with category names that mention the structure or landmark associated with the floor plan, e.g., *Angkor Wat*, *Plans of Guy's Hospital*, and *Blenheim Palace in art*. We iterate through the category tags and infer the name of the structure or landmark by removing prefixes like "Floor plans of", "Floor plan of", "Plans of", "Plans of the" or "Maps of", and suffixes like "in art". We then check if the structure is a scene of interest by checking if it is an instance of a predefined set of scene categories as in [34] (e.g., "religious building" or "castle"; the full list is included in the supplemental material). If it is indeed a scene of interest, we manually inspect the floor plan image to ensure it is a canonical floor plan (not a section plan or drawing too abstract where scene structure is hard to extract), before adding the floor plan image to our dataset. This process results in 10,842 floor plans from a total of 6,194 scenes.

## 3.2 Collecting Corresponding Photos

We use MegaScenes [34] as the primary source of photos because it contains a large number of in-the-wild photos that are already grouped by scenes. We take the intersection of scenes that are represented in MegaScenes with the scenes for which we have collected floor plans above. For some scenes, we also collect additional photos from YFCC100M [35] (for reasons explained in the next section). With this process, we end up with 1,474 scenes associated with a total of 766K photos and 2,942 floor plans.

## 3.3 Determining Correspondences

Once we have floor plans and sets of photographs from the same scenes, the next step is to annotate correspondences between floor plans and photos. However, manually annotating correspondences at this scale is an infeasible task. Instead, we use a two-step process: 1) automatic SfM reconstruction of the photo collection corresponding to each scene using COLMAP [22], and 2) manual alignment of the resulting point clouds with the floor plan for that scene. Given a set of images, COLMAP estimates a 3D camera pose for each image, as well as a sparse point cloud corresponding to keypoints in the photo collection. Aligning these point clouds with the floor plan thus directly yields correspondences between individual photos and the floor plan and is a substantially easier task than manually annotating correspondences. We describe each step below.

### 3.3.1 COLMAP Reconstruction from Photo Collections

We use default parameters for feature extraction and sparse reconstruction. For scenes with a small number of photos, we use exhaustive matching with default parameters. Running exhaustive matching on scenes with a large number of photos takes a prohibitive amount of time, so we instead use vocabulary tree matching with 40 nearest neighbors.

For some scenes, COLMAP on MegaScenes photos results in disjoint components and very sparse reconstructions. As such, we augment our photo collection with YFCC100M [35] for all scenes. Since YFCC100M photos are geotagged, for each scene, we download all photos that are within a 50-meter radius of that scene's GPS location (retrieved from Wikimedia Commons). We then use this augmented set of images to perform COLMAP reconstruction.

Finally, as a post-processing step, we run COLMAP's model merger on all pairs of reconstructed 3D components for each scene. This step attempts to merge any components with overlapping cameras or 3D points and results in more unified reconstructions (although many scenes will still have separate components, e.g., for the interior and an exterior of a building).

### 3.3.2 Manual Alignment of Point Clouds to Floor Plans

We devised a custom user interface that displays SfM point clouds and floor plans for a given scene and allows annotators to interactively apply transformations to each scene component's point cloud to align it to the floor plan. The interface displays a floor plan and a bird's-eye view of a point cloud. This viewpoint simplifies the matching process by limiting the floor plan and point cloud transformations

to only 2D translations, rotation, and scale. Once manually aligned, the transformation parameters mapping the point cloud to the floor plan, $T_{pc \to fp}$, are saved in a database as a transformation that maps points in the point cloud to the floor plan. In this work, we repeat this alignment for the two largest reconstructed point clouds for all the scenes.

Once we have these alignments, it is fairly straightforward to obtain sparse correspondences between the floor plan and the corresponding photos. We simply take each reconstructed point $X$ that is visible in a photo $i$ and project it into both the photo (using COLMAP-estimated camera parameters $T_i$) and the floor plan (using the manually estimated transformation $T_{pc \to fp}$), where $x_i$ and $x_{fp}$ are corresponding points:

$$x_{fp} = T_{pc \to fp} X \tag{1}$$
$$x_i = T_i X \tag{2}$$

We can apply a similar transformation to obtain the camera pose of each photo.

This yields our full dataset of floor plans, corresponding photos, correspondences between the two, and camera poses of the photos, which we report in Section 3.4. Note that there is a drop in number of scenes, floor plans, and photos from Section 3.2 because not every scene has a reconstruction and many reconstructions are sparse and thus not alignable. We also manually inspect floor plans and discard composite and ambiguous ones, where an image contains floor plans of many scenes and floor plans for multiple floors of a scene, respectively.

### 3.4 Dataset Statistics

The C3 dataset comprises 90K floor plan-photo image pairs derived from 597 scenes. These scenes span 648 unique floor plans and include 85K photos. The dataset also includes a camera pose for each photo and 153M total pixel-level correspondences. The number of correspondences per plan-photo pair ranges from 1 to 13,262, with an average of 1,711 correspondences per pair.

We split the dataset into train and test sets by scene, ensuring no scene-level overlap between them. We train on 479 scenes, which consists of 519 unique floor plans, 66K photos (and camera poses), and 120M correspondences. We test on 118 scenes, which contains 129 unique floor plans, 19K photos (camera poses), and 33M correspondences.

## 4 Evaluation on C3

We first detail the correspondence baselines and show that existing baselines from the literature struggle on the cross-view cross-modality correspondence task in Section 4.1. In Section 4.2, we share our approach and discuss our results and findings. We show camera pose estimation as a downstream application of our predicted correspondences in Section 4.3.

### 4.1 Baseline Performance

**Method.** We evaluate our dataset with a combination of sparse, semi-dense, and dense matching algorithms: SuperGlue [7], LoFTR [4], DINOv2 [2], DIFT [3], RoMa [5], and MASt3R [6]. We also evaluate on DUSt3R [1]. Since our goal is to find dense correspondences between images, we make adjustments to these methods, detailed below. We start with the matching methods and leave DUSt3R and MASt3R for last. Since SuperGlue produces sparse correspondences, we perform nearest neighbor interpolation to create a dense correspondence map. DINOv2 outputs patch-level features, so we upsample the features to full image resolution using bilinear interpolation and then compute pixel correspondence with cosine similarity. For LoFTR, DIFT, and RoMa, we can sample correspondences directly with pixel coordinates. We use LoFTR's `outdoor-ds` pre-trained model and RoMa with default `scale_factor` and `upsample_factor=(512, 512)`. With DUSt3R, we input floor plan as the reference image (that is, the image that defines the 3D coordinate frame) along with a photo. This way, DUSt3R's pointmap representation maps each photo pixel to a 3D point at location $(x, y, z)$ in the floor plan's coordinate frame. To obtain an actual pixel correspondence on the floor plan, we perform an orthographic projection $(x, y, z) \to (x, z)$, i.e., we drop the $y$-coordinate because the $y$-axis represents the up direction with respect to the coordinate frame of the first image (the floor plan in this case). With MASt3R, we provide the input images in the same order as DUSt3R.

| | RMSE ($\downarrow$) |
|---|---|
| SuperGlue | 0.4050 |
| LoFTR | 0.2901 |
| DINOv2 | 0.5338 |
| DIFT | 0.3036 |
| RoMa | 0.3308 |
| DUSt3R | 0.2925 |
| MASt3R | 0.4616 |
| Ours | **0.1919** |

Figure 2: Quantitative results for C3 test set (floor plan and photo pairs from scenes not used during training). Left: table of RMSE values (lower is better). Our method, trained on C3 training data, achieves a significant reduction in error. Middle: Percentage of Correct Keypoints (PCK) as a function of error threshold. Right: Precision-Recall curves generated by thresholding on predicted confidence or score for each method.

To ensure dense matches, we obtain the pixel correspondences from the prediction head without any follow-up filtering steps.

**Results.** We report quantitative results in Figure 2. The left table lists RMSE scores for each model. To standardize the RMSE calculation, we normalize all model outputs to a range of $[0, 1]$ (that is, the image dimensions are remapped to a unit square). The middle graph shows percent of correct keypoints (PCK), which measures the proportion of predicted correspondence points that fall within a certain threshold distance of the ground truth points. In our case, our distance metric is the Euclidean distance. The right graph displays Precision-Recall (PR) curves for the methods that output a confidence score associated with the correspondences, and we consider a prediction to be correct if its Euclidean distance from the ground truth is less than 0.05 units in normalized floor plan coordinates. We show qualitative comparisons in Figure 3. Unsurprisingly, all correspondence-based methods exhibit poor performance as they have not been trained on floor plan data. Although MASt3R—a network built on the DUSt3R model for matching tasks—might be expected to outperform DUSt3R, it actually shows higher error. One explanation could be that MASt3R is performing correspondence estimation, while DUSt3R is predicting scene structure and projecting it onto the 2D floor plan which is meaningfully closer to the solution.

## 4.2 Cross-View Cross-Modality Correspondence by Pointmap Prediction

While the baseline results were rather poor, we observe promising geometric structures in DUSt3R outputs; the model only needed to learn the 2D translations, rotation, and scale to align to the floor plan. We therefore leverage the strong geometric prior from the pre-trained DUSt3R model and fine-tune on our dataset, with some modifications. First, we split DUSt3R's Siamese encoders, which were designed to process two input images with visual overlap. Since our inputs—floor plans and photos—are from different domains, we reason that each encoder should separately learn the distributions of each individual domain. We also find we can treat this correspondence task as a pointmap prediction problem. As explained in Section 4.1, we can set up DUSt3R's pointmap representation to map 2D points in the photo to 3D points in the floor plan coordinate frame, then project back to 2D via an orthographic projection that discards the $y$- (or up-) coordinate. We also experiment with discarding the $z$-coordinate instead, but this empirically leads to slower model convergence. Finally, we observe model overfitting on floor plans during training. To improve model generalization, we perform photometric augmentations (color jitter) and geometric augmentations (cropping and rotation) on the floor plans.

**Training Setup.** We train our approach for 10 epochs which takes about 3 days with $8\times$A6000 40GB. We use the same hyperparameters as DUSt3R. We share more details regarding our setup in the supplementary material.

**Results.** Quantitatively, our model displays a 34% decrease in RMSE compared to the best performing baseline. We also observe a stronger PCK and PR performance for our model. Figure 3 shows that our model predicts the correspondences accurately, sometimes less noisily than the COLMAP

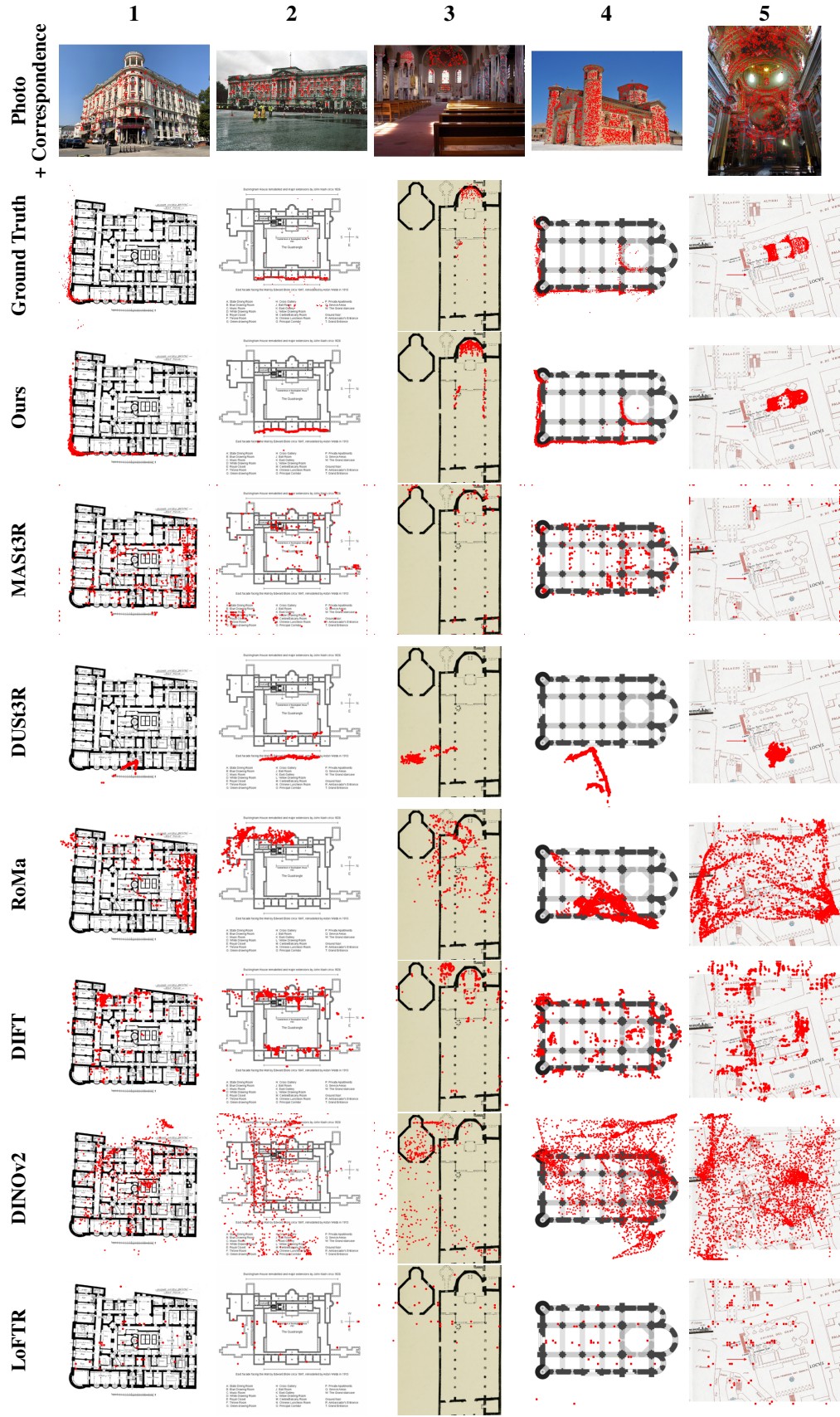

Figure 3: Qualitative results for C3 test set. Each red dot in the images represents a correspondence point.

| | | C3 R@ | | | | | | | |
|---|---|---|---|---|---|---|---|---|---|
| Method | Angular | | | | Positional | | | Angular +
Positional | |
| | 5° | 10° | 20° | 30° | 5% | 10% | 20% | 30°, 20% | |
| C3Po (ours) | 21.86 | 32.53 | 44.12 | 51.35 | 15.94 | 27.73 | 41.21 | 29.48 | |

Table 1: Camera pose estimation results on the C3 dataset, measured by recall under varying angular, positional, and both angular and positional thresholds. The positional thresholds represent a percentage of the floor plan's diagonal length.

reconstructions. We perform the Wilcoxon signed-rank test, a non-parametric paired test, between our error and each baseline and find all P-values to be less than 0.05.

### 4.3 Camera Pose Estimation

**Method.** We now explore the task of estimating the 2D camera pose of the ground-view photo in the floor plan coordinate frame—a challenging "you are here" photo location identification problem. We first estimate epipolar geometry via correspondences found with our method. To simulate the floor plan's orthographic projection, we use a large focal length ($10^7$ pixels). Given matched keypoints between the photo and plan, we compute an essential matrix using OpenCV's `findEssentialMat` function, then decompose the essential matrix into camera rotation and translation. To transform the camera rotation matrix from the XY plane to the floor plan coordinate frame, which lies on the XZ plane, we apply a $-90°$ rotation about the $x$-axis. We find that in our case an ambiguity can arise for photos that observe a near-planar structure (as vertical planes like walls yield points that lie on a *line* in the floor plan). Such cases can yield upside-down cameras on the wrong side of the observed surface. We detect such cases automatically based on the estimated rotation matrix and correct them by reflecting the camera across the best-fit plane formed by the correspondences. Finally, we estimate the camera center by computing the epipole in the floor plan image. The epipole in the floor plan corresponds to the camera location of the ground-view photo.

**Results.** We evaluate camera poses estimated from correspondences predicted by C3Po on the C3 dataset using a range of angular, positional, and combined angular-positional thresholds, as reported in Table 1. Positional thresholds are expressed as the Euclidean distance between estimated and ground-truth camera centers, normalized by the diagonal length of the floor plan.

## 5 Open Challenges

While our method demonstrates encouraging results, we show two categories of examples where our model could improve on.

### 5.1 Challenge 1: Photos with Minimal Context

This case refers to floor plan-photo pairs where the photo lacks contextual information of its global surroundings. These are examples that would be challenging even for humans to reason about in terms of the general location of the correspondences on the floor plans or camera pose. For example, Figure 4 (top row) shows a close-up shot of a door and a window. The second row figure is a photo of only an artwork. In both instances, our model makes plausible predictions; for example, the door photo is mapped to one of the doors on the floor plan and is along the exterior wall of the scene. However, the answer is wrong due to lack of context. Future work could attempt to resolve this issue by, for instance, predicting distributions of correspondences, rather than regressing to a specific unimodal answer.

### 5.2 Challenge 2: Structural Symmetry

This challenge involves scenes with structural symmetry. Although there are subtle cues on the floor plan that can often help disambiguate scenes that feature symmetries (domes, similar walls or

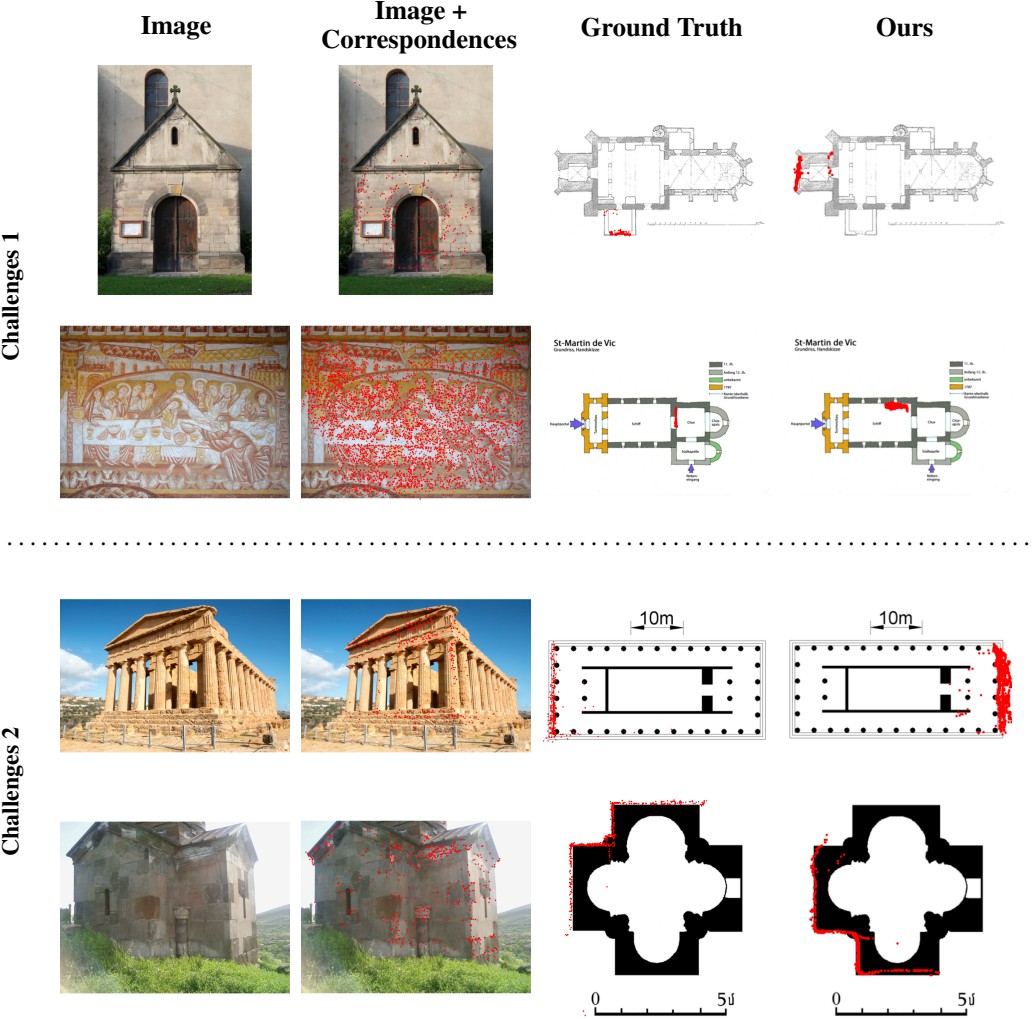

Figure 4: We share two categories of data that our model struggles on. Challenge 1, top two rows, are cases where the photo provides minimal context clues of where it could be on the floor plan. Challenge 2, bottom two rows, are scenes that exhibit structural symmetry, where multiple correspondence alignments would seem plausible.

hallways, etc.), they are difficult to identify from the photos. Figure 4 (third row) shows a photo of a temple viewed from the left side of the floor plan and the last row shows a photo of a church viewed from the top left corner. Again, our model predicts plausible correspondence configurations that are consistent with the scene geometry in both cases, and again, perhaps a more distributional approach to prediction, e.g., with diffusion models, would be more appropriate in such cases.

## 6 Conclusion

In this paper, we present C3, the first cross-view, cross-modality correspondence dataset. We first source floor plans and photos of scenes from the Internet and then determine their correspondences by running COLMAP followed by carefully aligning the reconstructed point clouds to the floor plans. We show that existing correspondence methods fail to accurately establish matches between floor plans and images. We propose to frame matching as a DUSt3R pointmap prediction task, and this approach outperforms the best performing baseline by 34%. We further showcase the utility of these correspondences for camera pose estimation as a downstream task. We also highlight structured failure modes for future research directions.

In addition to the open challenges we observe, our dataset could be used to enable a number of other cross-modal tasks. For instance: (1) given an image and a floor plan, localize the image on the plan (i.e., camera-to-plan relative pose), (2) given a floor plan and a camera, generate an image (i.e., floor plan-conditioned image generation), and (3) given an image, generate a complete floor plan of the structure pictured (image-to-floor-plan generation). In general, we hope that having access to quality data can help spur progress on problems involving jointly reasoning about the kinds of global, abstracted structure available in a floor plan and the local structure pictured in a photo.

**Societal Impacts.** Through our dataset and approach, we hope to enable researchers to develop more accurate and robust methods in areas including 3D vision, image generation, and robotics. We do not anticipate negative societal concerns.

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
