# OpenReview forum: "C3Po: Cross-View Cross-Modality Correspondence by Pointmap Prediction"
_NeurIPS.cc/2025/Datasets_and_Benchmarks_Track — NeurIPS 2025 Datasets and Benchmarks Track poster_

### Official Review · Reviewer_kV54 · 2025-06-26

**Rating:** 4
**Confidence:** 3

**Summary:**

This paper introduces a novel and compelling benchmark, C3Po, which consists of floor plan–photo pairs with pixel-level correspondences. Building upon this benchmark, the authors evaluate representative correspondence methods, providing valuable insights into the limitations and bottlenecks of current models. They also finetune an existing model, achieving state-of-the-art performance on the proposed dataset.

**Dataset Code Accessibility:**

Yes

**Ethical Considerations:**

No, there are no or only very minor ethics concerns

**Final Justification:**

I read the rebuttal, and other reviews. Most of my concerns are addressed, except the first (main) one. Considering this paper proposes an interesting topic, and the representation and experiments need to be further improved, I raised my score to Borderline accpet 4 but not accept 5.

**Limitations Weaknesses:**

1. DUSt3R performs poorly on this task, due to the lack of similar examples in its training data. The authors address this by finetuning the model, leading to a significant improvement. However, this raises a key concern:
Does the finetuned model specialize only in the C3Po dataset, or can it generalize to broader 3D vision tasks? It would be helpful to evaluate the finetuned model on more general matching and reconstruction tasks (e.g., those used to benchmark DUSt3R) to assess its broader applicability.

2. Regarding the “Structural Symmetry” challenge mentioned in the paper: How is matching accuracy determined in cases where the floor plan is perfectly symmetrical? In such cases, is there a designated reference coordinate system, or are multiple correspondences considered equally valid ground truth? Clarification on this evaluation protocol would strengthen the paper.

3. (Line 133): The authors state: “This process results in 10,842 floor plans from a total of 6,194 scenes.” It would be helpful to provide more detailed statistics, such as: The number of raw samples before filtering. The number of samples discarded or retained at each stage.

4. (Line 146): The paper mentions the “manual alignment of the resulting point clouds with the floor plan.” While the supplementary material includes a UI screenshot, it is unclear how this manual process ensures pixel-level accuracy. What guidelines or quality checks were applied during annotation? How did annotators ensure alignment precision?

5. The experiments primarily use RMSE as the evaluation metric. Would it be possible to include additional metrics, such as the percentage of pixels below a certain error threshold? This could provide a more nuanced and comprehensive understanding of model performance.

**Strengths Contributions:**

The topic and problem setting introduced by this benchmark are both novel and interesting.

The supplementary materials are thorough and provide detailed insights into the data collection and annotation processes.

---

> ### Author Rebuttal · Authors · 2025-07-31
>
> We thank the reviewer for your constructive and detailed feedback. Below we respond to your concerns.
>
> ## Evaluating C3Po on More Tasks:
> This is a good question. Due to the tight rebuttal timeline, we plan to evaluate C3Po on other tasks, including visual localization, for the camera-ready.
>
> ## Structural Ambiguity:
> Even when the floor plan is perfectly symmetrical, we take the human labeled data as groundtruth. We acknowledge that the model may produce an equivalent set of correspondences given the symmetries but will still be penalized. We also recognize that the ambiguities represent an interesting challenge, as we mention in Section 5.2 ("Structural Ambiguity"), and that future work can attempt to tackle this problem, perhaps with a distributional approach. In the meantime, to help separate the role of various challenges in our evaluation, we have manually inspected and collected a small set of 14 unambiguous floor plan and image pairs. We note that finding all unambiguous structures in all plan-image pairs is a challenging and laborious manual task, which is why we only evaluate on the small set. We observe that the RMSE errors for baselines on the unambiguous case are generally consistent with those from the full dataset. The overall relative baseline performance ranking among methods is also generally preserved.
>
> |           | RMSE (↓) |   |   |   |
> |-----------|----------|---|---|---|
> | SuperGlue | 0.3358   |   |   |   |
> | LoFTR     | 0.2877   |   |   |   |
> | DINOv2    | 0.5938   |   |   |   |
> | DIFT      | 0.3400   |   |   |   |
> | DUSt3R    | 0.2536   |   |   |   |
> | MASt3R    | 0.5031   |   |   |   |
> | RoMa      | 0.3613   |   |   |   |
> | Ours      | **0.0295**   |   |   |   |
>
> ## More Detailed Statistics on Data Collection:
> Below, we share the number of samples at every stage of the data creation process. While some of these statistics are already included in the submitted paper, we have provided them below again for clarity and convenience.
>
> Raw floor plans: 20K floor plans
>
> After filtering floor plans: 6,194 scenes; 10,842 floor plans
>
> Raw photos (MegaScenes): 430K scenes; 9M photos
>
> After filtering photos: 1,474 scenes; 2,942 floor plans; 766K photos
>
> Flickr photos: 1,086 scenes; 241K photos
>
> Final dataset: 574 scenes; 623 floor plans; 86k photos
>
> ## Ensuring the Accuracy of the Manual Alignment Process:
> During the manual annotation process, we ensure that the majority of the points in each point cloud should overlap with the lines on the floor plans, indicating an alignment in structure. To ensure the reliability of the alignments, we also ask 4 different people to label 10 randomly sampled point cloud-floor plan pairs. We then compare their labels with the dataset’s existing labels and find that on average the aligned points vary by less than 5% of max (floorplan width, floorplan height). This indicates minimal variation between the user-provided labels and the ones in our dataset and that the alignment process is reliable. Additionally, we would like to mention that since floor plans are abstract representations, obtaining a “real” groundtruth is inherently challenging. This is why we relied on manual alignment as the best available approach for this task.
>
> ## Additional Evaluation Metrics:
> In addition to using RMSE as an evaluation metric, our paper also reported
> 1. Lines 218-221: “percent of correct keypoints (PCK), which measures the proportion of predicted correspondence points that fall within a certain threshold distance of the ground truth points. In our case, our distance metric is the Euclidean distance.”
>
> 2. Lines 221-223: “Precision-Recall (PR) curves for the methods that output a confidence score associated with the correspondences, and we consider a prediction to be correct if its Euclidean distance from the groundtruth is less than 0.05 units in normalized floor plan coordinates."
>
> 3. Figure 2: The PCK and PR Curves graphs.
>
> Thank you again for helping us improve this work!

---

> > ### Comment · Reviewer_kV54 · 2025-08-05
> >
> > I read the rebuttal, and other reviews. Most of my concerns are addressed, except the first (main) one. Considering this paper proposes an interesting topic, and the representation and experiments need to be further improved, I raised my score to Borderline accpet 4 but not accept 5.

---

### Official Review · Reviewer_7rcu · 2025-06-28

**Rating:** 4
**Confidence:** 3

**Summary:**

This paper addresses a challenging problem that has been largely overlooked by previous research: predicting correspondences between ground-level photographs and floor plans. To facilitate this study, the authors introduce a new dataset named C3, constructed by first reconstructing a series of scenes in 3D using Internet photo collections through structure-from-motion techniques. Experimental results demonstrate the effectiveness of the proposed dataset. In addition, the paper identifies several open challenges in cross-modal geometric reasoning.

**Dataset Code Accessibility:**

Yes

**Ethical Considerations:**

No, there are no or only very minor ethics concerns

**Final Justification:**

This paper tackles a challenging and relatively underexplored problem—predicting correspondences between ground-level photographs and floor plans. To support this research, the authors introduce a new dataset, C3. The proposed method appears technically sound; however, its potential for real-world applications remains to be fully explored. Overall, I consider this a borderline accept.

**Limitations Weaknesses:**

1. Could this be extended to multi-view sparse inputs instead of a single view? Incorporating multiple views would provide richer information, which may help address the challenges discussed in Section 5. VGGT[1] maybe a good baseline.

2. The authors may consider whether textual annotations in floor plans should be removed before model input, as they might influence performance or lead to unintended shortcuts.

[1] Wang J, Chen M, Karaev N, et al. Vggt: Visual geometry grounded transformer[C]//Proceedings of the Computer Vision and Pattern Recognition Conference. 2025: 5294-5306.

**Strengths Contributions:**

1. The problem statement is clear and easy to understand.

2. The motivation is reasonable. Addressing the task of localizing oneself given only a map and a few sparse views is important and has practical relevance.

3. The experimental results are impressive and demonstrate the effectiveness of the proposed method.

---

> ### Author Rebuttal · Authors · 2025-07-31
>
> Thank you for your insightful and constructive suggestions. Below we respond to your concerns.
>
> ## Extending to Multi-View Inputs:
> This is a great suggestion. We agree that extending our method to handle multi-view sparse inputs is indeed a promising direction and could help mitigate the challenges we discussed in Section 5 by providing additional spatial context. However, we would like to note that VGGT was not published at the time of our submission, and due to the tight timeline of the rebuttal period, we plan to explore this direction as future work.
>
> ## Effect of Removing Textual Annotations on Floor Plans:
> To assess the impact of textual annotations, we perform a small pilot experiment where we use one of the plan-photo pair in our GitHub repository (C3Po/demo/images/plans/119882_plan.jpg and C3Po/demo/images/photos/119882_photo.jpg), which contains a variety of texts, including a title, legend, compass, scale bar, and room names. We removed all textual annotations using white overlays and compared the predicted correspondences with those from the original floor plan with text. We find that the removal of text annotations has negligible qualitative and quantitative differences in the predictions. We report the RMSE errors for both the annotated and non-annotated plans. We acknowledge this result is not definitive, but it suggests that text annotations play a small role in correspondence prediction. We plan to add additional experiments and  qualitative results in the camera-ready.
>
> |                                               | RMSE (↓)  |   |   |   |
> |-----------------------------------------------|--------|---|---|---|
> | With text annotation on floor plan (original) | 0.0261 |   |   |   |
> | Without text annotations on floor plan        | 0.0274 |   |   |   |
> |                                               |        |   |   |   |
>
> Thank you again for helping us improve this work!

---

> > ### Comment · Reviewer_7rcu · 2025-08-05
> >
> > Thank you to the authors for the response. I have no further questions and maintain my score.

---

### Official Review · Reviewer_SoQT · 2025-07-01

**Rating:** 3
**Confidence:** 3

**Summary:**

In this manuscript, the authors propose a new dataset, named C3, with pixel-level correspondences between the photos and floor plans. The dataset is collected by performing structure from motion (sfm) with colmap on photos to obtain the point clouds and then manual alignment between the point clouds with floor plans.

The C3 dataset consists of 91K paired floor plans and photo images across 574 scenes with 155M pixel-level correspondences.

Experiments demonstrate that existing methods trained only on photos report poor performance with large errors. After finetuning the Dust3R on the proposed C3 dataset, the performance has been improved significantly with 34% lower errors

**Dataset Code Accessibility:**

Yes

**Ethical Considerations:**

No, there are no or only very minor ethics concerns

**Final Justification:**

I would like to thank the authors for their update. After reading the authors' response, I still have concerns about the advantages of this dataset in real applications. Therefore, I keep my initial rating.

**Limitations Weaknesses:**

1.	My major concern is the usage of the proposed dataset. In the manuscript, the authors say that the pixel-level correspondences could be used for the localization of robotics. However, are pixel-level correspondences necessary for this task? As a matter of fact, for most cases, only rough localization results are needed and the high-level structure information is sufficient. If more precise localization is needed, the authors need to prove that in the experiments.

2.	In the manuscript, the authors mention that alignment between the photos and floor plans are performed manually, which means the groundtruth are not real groundtruth and how to measure the errors in the manual alignment is a problem.

3.	Matching ambiguity is another problem. For example, a photo of s single wall could be matched with both sides of the wall on the floor plan, leading to potential errors. How to solve this problem?

**Strengths Contributions:**

1.	The multi-modality C3 dataset. The C3 dataset contains multimodality data with pixel-level correspondences between the photos and floor plans.  As mentioned in the manuscript, this dataset could be used for robotic navigation (although I personally pixel-level correspondences are not necessary).

2.	Experiments show that existing features and matching methods fail to report satisfying performance on this dataset. It makes sense because these methods are trained only on normal images. This also explains why the finetuned dust3r works much better after fine-tuning on the proposed dataset.

3.	The paper is well-organized and easy to read.

---

> ### Author Rebuttal · Authors · 2025-07-31
>
> Thank you for your thoughtful comments. We respond to your concerns below.
>
> ## Pixel-Level Correspondence for Localization and Usefulness of This Dataset and Task:
> We would like to point out that pixel-level correspondence is a standard and widely-used method for localization. Regardless of whether the desired localization is rough or precise, the task still requires estimating both a location and orientation and this is where correspondences are helpful. Also, note that in addition to pixel correspondences, we plan to release camera poses as well. This allows for alternative approaches to leverage our dataset, like to learn to regress directly to camera poses, without first predicting pixel correspondences. Note that correspondence and camera pose estimation are useful not just for localization but also for tasks like 3D reconstruction.
>
> ## Manual Alignment Process:
> We agree with the reviewer that ensuring the accuracy of the manual alignments is crucial. To ensure the reliability of the alignments, we perform an evaluation where we ask 4 different people to label 10 randomly sampled point cloud-floor plan pairs. We then compare their labels with the dataset’s existing labels and find that on average the aligned points vary by less than 5% of max (floorplan width, floorplan height). This indicates minimal variation between the user-provided labels and the ones in our dataset and suggests that the alignment process is reliable. Additionally, since floor plans are abstract representations, obtaining a “real” groundtruth is inherently challenging, and manual alignment is the best available approach for this task.
>
> ## Addressing Structural Ambiguity:
> We agree with the reviewer that ambiguities represent an interesting challenge, as we mention in Section 5.2 ("Structural Ambiguity"). We believe that this is indeed one of the challenges that our new benchmark raises, and that future work can attempt to tackle this problem, perhaps with a distributional approach. In the meantime, to help separate the role of various challenges in our evaluation, we have manually inspected and collected a small set of 14 unambiguous floor plan and image pairs. We note that finding all unambiguous structures in all plan-image pairs is a challenging and laborious manual task, which is why we only evaluate on the small set. We observe that the RMSE errors for baselines on the unambiguous case are generally consistent with those from the full dataset. The overall relative baseline performance ranking among methods is also generally preserved.
>
> |           | RMSE (↓) |   |   |   |
> |-----------|----------|---|---|---|
> | SuperGlue | 0.3358   |   |   |   |
> | LoFTR     | 0.2877   |   |   |   |
> | DINOv2    | 0.5938   |   |   |   |
> | DIFT      | 0.3400   |   |   |   |
> | DUSt3R    | 0.2536   |   |   |   |
> | MASt3R    | 0.5031   |   |   |   |
> | RoMa      | 0.3613   |   |   |   |
> | Ours      | **0.0295**   |   |   |   |
>
> Thank you again for helping us improve this work!

---

> > ### Comment · Reviewer_SoQT · 2025-08-06
> >
> > Many thanks for the update. I still have the opinion that for coarse localization pixel-wise correspondences are not necessary. After reading other reviewers' comments, I also agree that the scale is also limited. Therefore, I keep my initial rating.

---

### Official Review · Reviewer_7iJD · 2025-07-05

**Rating:** 3
**Confidence:** 4

**Summary:**

The C3 dataset is a cross-view cross-modality correspondence dataset designed to facilitate research in 2D-3D matching tasks. Such dataset offers significant research value and practical significance in addressing cross-view cross-modality correspondence tasks. It provides researchers with abundant data resources and benchmarking tools. However, it also has certain limitations, such as a limited dataset scale, incomplete modalities, and potential annotation errors. By addressing these shortcomings, the dataset can better meet the needs of researchers and further advance the development of cross-view cross-modality correspondence research.

**Dataset Code Accessibility:**

Yes

**Ethical Considerations:**

No, there are no or only very minor ethics concerns

**Final Justification:**

I prefer to keep my rate concerning the expectation of this dataset paper submission

**Limitations Weaknesses:**

- *Limited Dataset Scale*: While the dataset covers a variety of scene categories, the number of scenes may still be relatively limited. This could restrict the generalization and robustness of algorithms trained and tested on the dataset. Expanding the dataset to include more scenes would further enhance its value.
- *Incomplete Modalities*: The dataset primarily focuses on correspondences between 2D images and 3D floor plans. However, there may be a lack of diversity in other modalities, such as depth images or dense 3D point clouds. In practical applications, matching across multiple modalities is a common challenge. Enriching the dataset with additional modalities would better meet the demands of cross-modal correspondence research.
- *Limited Benchmarking Methods*: While the paper provides benchmarking results for the C3Po model, comparisons with other state-of-the-art methods are relatively limited. Including benchmarking results from more algorithms would better demonstrate the dataset's effectiveness and provide researchers with a more comprehensive reference.

**Strengths Contributions:**

- *Research Value*: The C3 dataset addresses the critical issue of establishing correspondences between 2D images and 3D floor plans in cross-view and cross-modality scenarios. It holds substantial research value for applications such as 3D reconstruction, visual localization, and semantic mapping. The dataset provides researchers with abundant data resources and benchmarking tools to advance the development of related algorithms.
- *Comparably Rich Scene Categories*: The dataset encompasses a wide range of predefined scene categories, including amphitheaters, architectural structures, basilicas, buildings, castles, cathedrals, and more. This diversity enables researchers to evaluate and compare the performance of algorithms across different architectural styles and scene types, offering comprehensive support for research in various scenarios.
- *Ready-to-use Correspondence*: Each scene in the dataset includes correspondences between floor plans and photos. The correspondences are derived from high-quality pointmaps predicted by the C3Po model, accompanied by confidence scores. Researchers can utilize this information to analyze the reliability of correspondences and develop more robust matching algorithms.
- *User-Friendly Interface*: A custom user interface is provided to align point clouds with floor plans. This interface offers interactive functionalities such as 2D translation, rotation, and scaling adjustments for floor plans and point clouds. It also includes additional features like Google Earth links, 3D mesh views, and scene images, making it easier for users to understand and align the geometric relationships within scenes. These tools enhance the usability and practicality of the dataset.
- *Comprehensive Benchmarking Results*: The paper presents detailed benchmarking results for the C3Po model on the C3 dataset, including RMSE and PR curves. These results demonstrate the superiority of the C3Po model and provide a reference for researchers using the dataset to evaluate their algorithms.

---

> ### Author Rebuttal · Authors · 2025-07-30
>
> Thank you for your thoughtful reviews. We respond to your concerns below.
>
> ## Limited Dataset Scale:
> We recognize the reviewer’s concern about the current scale of the dataset. We would like to highlight that our dataset already encompasses a diverse range of scene categories. Moving forward, we plan to periodically update our dataset to include more scenes and correspondences, further improving the generalization and robustness of models trained on this dataset. For example, an updated version of MegaScenes containing 30% more scenes is expected to increase our dataset and this is only accounting for photos from MegaScenes.
>
> ## Incomplete Modalities:
> We focus on floorplans because floorplans provide unique challenges and opportunities. In terms of challenges, floorplans are abstract representations of the 3D space, and we wanted to test if 3D reconstruction models can bridge the gap between photographs and these abstract representations. In terms of opportunities, registering photos to floorplans can enable the registration of far away or non-overlapping photos to each other (one example is indoor and outdoor photos), which is a challenge for 3D reconstruction techniques. Note that while we focus on floorplans for this benchmark, similar architectural choices could also be useful for other modalities like aerial photos.
>
> ## Limited Benchmarking Methods:
> As requested, we evaluate our dataset on an additional state-of-the-art correspondence model, namely RoMa [1]. We observe that our method outperforms RoMa, most likely because the appearance changes from the vast viewpoint and modality/style differences between the photo and floor plan. Since we can only share the quantitative results in the rebuttal stage, we will update PCK and PR Curve graphs and quantitative results in the camera-ready. We will also explore other baselines for the camera-ready. If the reviewer has particular baselines in mind, we are happy to add them.
>
> |      | RMSE (↓) |   |   |   |
> |------|---------|---|---|---|
> | RoMa | 0.3308  |   |   |   |
> | Ours | **0.1919**  |   |   |   |
> |      |         |   |   |   |
>
> Thank you again for helping us improve this work!
>
> [1] Edstedt J, Sun Q, Bokman G, Wadenback M, Felsberg M. RoMa: Robust Dense Feature Matching. CVPR 2024.

---

> > ### Comment · Reviewer_7iJD · 2025-08-05
> >
> > The updated benchmarking efforts provide more insight, which is plausible. I'm still concerned about the data scale and modality given the fact that this is a dataset paper submission. So I prefer to keep the original rate in general.

---

### Note · Authors · 2025-08-15

## Limited data size:
We agree that more samples per scene category could enhance generalizability. We plan to periodically increase the size of our dataset. For example, an updated version of MegaScenes containing 30% more scenes is expected to increase our dataset. We also plan to recover floor plans from scenes initially discarded as those scenes are not present in MegaScenes and pair them with ground level photos directly from WikiCommons. Our current results already demonstrate consistent and robust performance across a diverse range of scene categories. Since the photos we collected are from the Internet, varied and challenging conditions are also covered, including differences in lighting, viewpoints, etc. We believe that the dataset and method contributions will be valuable to 3D researchers, even if the dataset is currently not exhaustive in per-category instances.
## Incomplete modalities:
While additional modalities could expand the potential applications of our dataset, finding correspondences between floor plans and ground-level photos is an underexplored challenge in cross-view cross-domain 3D vision. Our work addresses this challenge directly at scale for the first time. Our results show that our model outperforms existing correspondence models. By effectively registering photos to floor plans, we can register far away or non-overlapping photos to each other (indoor and outdoor photos for example), addressing a major limitation for current 3D reconstruction techniques.
## Pixel-wise correspondences for localization:
We emphasize that pixel-level correspondence is one of the most common approaches for localization. We also plan to release camera poses in addition to pixel correspondences, which would enable applications beyond localization like 3D reconstruction.
## Ambiguous correspondences:
We acknowledge that in the current benchmark, a model may produce an equivalent set of matches given the symmetries but will still be penalized. We believe that this presents an interesting challenge for future work, as we mention in Section 5.2 ("Structural Ambiguity"), perhaps addressable with a distributional approach. Meanwhile, to help separate the role of various challenges in our evaluation, we have manually collected a small set of 14 unambiguous plan-photo pairs. Baseline RMSE results on this subset are generally consistent with those on the full dataset, and the overall relative performance ranking among methods is also generally preserved.

---

### Decision · Program_Chairs · 2025-09-18

**Decision:**

Accept (poster)

**Comment:**

The submission presents the  C3 dataset  -  a cross-view cross-modality correspondence dataset for  2D-3D matching tasks. The dataset has novel features, and addresses an important practical problem. The reviewer option is split.

The merits of the paper are very well summarized in the quality review of Reviewer 7iJD, who list five strengths and three weaknesses. We agree with the lists,  but we do not agree with his assessment of the Weakness: 1. limited dataset scale, 2. incomplete modalities and 3. limited benchmarking. All three can be described "I would have liked more content", but: ad 2. the 2D-3D problem is challenging enough, ad 3. the benchmarking presented is sufficient to justify the need and value of the data. The other negative reviewers mentions three issues, but they are well covered in the rebuttal.

The strengths of the contributions outweigh the essentially minor limitations.